# Ivermectin as a Broad-Spectrum Host-Directed Antiviral: The Real Deal?

**DOI:** 10.3390/cells9092100

**Published:** 2020-09-15

**Authors:** David A. Jans, Kylie M. Wagstaff

**Affiliations:** 1Nuclear Signaling Lab., Department of Biochemistry and Molecular Biology, Biomedicine Discovery Institute, Monash University, Monash, Victoria 3800, Australia; 2Cancer Targeting and Nuclear Therapeutics Lab., Department of Biochemistry and Molecular Biology, Biomedicine Discovery Institute, Monash University, Monash, Victoria 3800, Australia; Kylie.Wagstaff@monash.edu

**Keywords:** ivermectin, antiviral, SARS-CoV-2, COVID-19, flavivirus, dengue virus, Zika virus

## Abstract

The small molecule macrocyclic lactone ivermectin, approved by the US Food and Drug Administration for parasitic infections, has received renewed attention in the last eight years due to its apparent exciting potential as an antiviral. It was identified in a high-throughput chemical screen as inhibiting recognition of the nuclear localizing Human Immunodeficiency Virus-1 (HIV-1) integrase protein by the host heterodimeric importin (IMP) α/β1 complex, and has since been shown to bind directly to IMPα to induce conformational changes that prevent its normal function in mediating nuclear import of key viral and host proteins. Excitingly, cell culture experiments show robust antiviral action towards HIV-1, dengue virus (DENV), Zika virus, West Nile virus, Venezuelan equine encephalitis virus, Chikungunya virus, Pseudorabies virus, adenovirus, and SARS-CoV-2 (COVID-19). Phase III human clinical trials have been completed for DENV, with >50 trials currently in progress worldwide for SARS-CoV-2. This mini-review discusses the case for ivermectin as a host-directed broad-spectrum antiviral agent for a range of viruses, including SARS-CoV-2.

## 1. Introduction

The 2015 Nobel Prize for medicine recognizes the seminal contribution of Campbell and Ōmura in terms of the “wonder drug” ivermectin, a macrocyclic lactone 22,23-dihydroavermectin B produced by the bacterium *Streptomyces avermitilis* [1], as a novel therapeutic against “infections caused by roundworm parasites”; this was alongside Tu Youyou for her seminal work on artemisinin and malaria [2]. Discovered in 1975, ivermectin was marketed successfully from 1981 for parasitic infection indications in animals, and then approved for human use for activity against onchocerciasis (river blindness) in 1987. It has since been used successfully to treat a number of human parasitic worm infestations causing river blindness/filariasis, strongyloidiasis/ascariasis, ectoparasites causing scabies, pediculosis and rosacea [1,3]. More recent applications include to control insect mediators of infection, such as malaria [1,3,4,5]. Ivermectin is on the World Health Organization’s Model List of Essential Medicines [6].

From 2012 onwards, there have multiple reports that ivermectin has antiviral properties [4,5,7,8,9,10,11,12,13,14,15,16,17] towards a growing number of RNA viruses, including human immunodeficiency virus (HIV)-1, influenza, flaviruses such as dengue virus (DENV) and Zika virus (ZIKV) and, most notably, SARS-CoV-2 (COVID-19) [17]. Evidence for activity against DNA viruses is more limited, but encompasses Pseudorabies, polyoma and adenoviruses [18,19,20]. The basis of ivermectin’s broadspectrum antiviral activity appears to relate to the fact that ivermectin binds to, and inhibits, the nuclear transport role of the host importin α (IMPα) protein [11,18,20], which is known to mediate nuclear import of various viral proteins and key host factors, but other possible antiviral actions of ivermectin have been proposed (e.g., [12]), including in the case of SARS-CoV-2 (e.g., [21,22]). This mini-review will summarise the weight of evidence for ivermectin’s broad-spectrum antiviral activity and the basis of its IMPα-directed activity in light of the possibility that ivermectin could be a critically useful antiviral in the current SARS-CoV-2 crisis [6,17].

## 2. Ivermectin as a US Food and Drug Administration-Approved Anti-Parasitic Agent

It is difficult to overestimate the impact of ivermectin as a therapeutic agent to control various parasitic diseases [1,2,3,4,5,6]. It is administered as a single oral yearly dose (e.g., 150 or 200 μg/kg, respectively) to treat onchocerciasis and strongyloidiasis. Lymphatic filariasis is similarly treated in endemic areas with a once-yearly dose (300–400 μg/kg), or alternatively bi-yearly dosing (150–200 μg/kg) [23]. Ivermectin’s documented antiparasitic mode of action is through potentiating GABA-mediated neurotransmission, and by binding to invertebrate glutamate-gated Cl^−^ channels to effect parasite paralysis and death [24]. Selectivity comes from the fact that ivermectin does not readily penetrate the central nervous system (CNS) of mammals, where GABA functions as a neurotransmitter [24].

Doses up to 2000 µg/kg are well tolerated in patients with parasitic infections [23,25], with analysis of the first 11 years of mass global ivermectin (Mectizan) administration indicating a cumulative incidence of one serious adverse side effect case per million [4,26]. Similarly, although drug resistance can occur in animals, no resistance in humans has yet been confirmed in over 25 years. Based on this weight of evidence, ivermectin is unquestionably a safe, potent antiparasitic agent likely to be used as such long into the future [1,4].

## 3. Ivermectin as an IMPα Targeting Agent with Antiviral Activity

Transport into and out of the nucleus is central to eukaryotic cell and tissue function, with a key role to play in viral infection, where a common strategy used by viruses is to antagonize the cellular antiviral response [14,27]. The targeting signal-dependent mediators of this transport are the members of the IMP superfamily of proteins, of which there are multiple α and β forms [14,27]. The pathway mediated by the IMPα/β1 heterodimer is the best characterized pathway by which host proteins, including members of the signal transducers and activators of transcription (STATs) and nuclear factor kappa-light-chain-enhancer of activated B cells (NF-κB) transcription factor families, enter the nucleus through nuclear envelope-embedded nuclear pores. A large number of viral proteins (e.g., [27,28]) also use this pathway (see Figure 1), where IMPα within the IMPα/β1 heterodimer performs the adaptor role of specific targeting signal recognition, while IMPβ1 performs the main nuclear roles of binding to/translocation through the nuclear pores, and release of the nuclear import cargo within the nucleus (Figure 1) [27].

The importance of nuclear targeting of viral proteins to the nucleus in the infectious cycle has been demonstrated for a number of viruses. Mutagenic analyses, for example, show that specific recognition by IMPα is critical to nuclear localization of various viral proteins, such as DENV non-structural protein (NS) 5 [30]; significantly, DENV, which shows the same reduced interaction of NS5 with IMPα is severely attenuated, underlining the importance of the NS5-IMPα interaction for dengue infection. As has since been shown using a range of different small molecules, the critical importance of this interaction to dengue infection is the basis for the fact that multiple distinct small molecules that disrupt IMPα recognition of dengue NS5 are able to limit dengue infection [7,8,11,29,31]. In the case of ivermectin, this activity extends to a large number of different viruses (see below) [7,8,9,10,11,12,13,14,15,16,17], including SARS-CoV-2 [18]. Which SARS-CoV-2 proteins may access the nucleus in infected cells has not been examined in detail but, in terms of related coronaviruses, ORF6 (Open Reading Frame 6) protein from SARS-CoV-1 has been shown to bind IMPα [32], and ORF4b from MERS-CoV (Middle Eastern Respiratory Syndrome Coronavirus) is known to access the nucleus in NLS-dependent fashion [33]. Ongoing research will establish which of the SARS-CoV-2 ORFs may play comparable roles, and be potential targets of the impact of ivermectin on IMPα.

We identified ivermectin in 2011 in a proof-of-principle, high-throughput screen using recombinantly-expressed proteins and a 1200 compound library for inhibitors of HIV-1 Integrase (IN) recognition by IMPα/β1 [34]; specific inhibitors targeting IMPα/β1 directly (such as ivermectin) and not IN (such as budenoside) were identified using a nested counterscreen strategy [34,35]. Of several compounds subsequently confirmed to be active against IMPα/β1 and possess antiviral activity as a consequence [7,14,29,36], ivermectin has been the best characterized in this regard, and shown to have broad-spectrum activities, a number of which are summarized in Table 1. It was initially shown to inhibit nuclear import not only of IN, but also of simian virus SV40 large tumour antigen (T-ag) and other IMPα/β1-dependent (but not IMPβ1-dependent) cargoes, consistent with the idea that IMPα (not IN) is the direct target [34,35]. Subsequent work has confirmed this, with ivermectin’s ability to inhibit the nuclear accumulation of various different host, including NF-kB p65 [37] and viral proteins demonstrated in transfected and infected cell systems (see Table 1) [14,34]. Ivermectin’s ability to inhibit binding of IMPα to the viral proteins NS5 and T-ag has also been confirmed in a cellular context using the biomolecular fluorescence complementation technique [11].

Although targeting of IMPα by ivermectin was clearly supported by many years of research (see also below), direct binding to IMPα was only recently formally demonstrated using a set of biophysical techniques, including thermostability, analytical ultracentrifugation, and circular dichroism (CD) [11]. Importantly, the CD/thermostability studies indicate that binding of ivermectin by IMPα induces a structural change, which is likely the basis of IMPα’s inability to bind viral nuclear import cargoes. Strikingly, the structural change also appears to impair heterodimerisation of IMPα with IMPβ1 [11]; IMPα alone cannot mediate nuclear import, only within the heterodimer with IMPβ1. Thus, ivermectin inhibits nuclear import not only by preventing signal recognition by IMPα, but also by ensuring that the IMPα/β1 complex essential to mediate subsequent transport through the nuclear pore is prevented from forming. Interestingly, GW5074 (see Table 1) appears to have a close to identical mechanism to that for ivermectin in binding to IMPα to induce structural changes that prevent both cargo recognition and specific interaction with IMPβ1 [29]; there is no such data for GSP (see Table 1), which is believed to impact the NLS-binding site of IMPα and prevent cargo recognition more directly (manuscript in preparation).

## 4. Ivermectin as an Antiviral

Consistent with the fact that many viruses are known to rely on IMPα/β1-dependent nuclear import of specific viral proteins for robust infection [14,27,28], ivermectin has been confirmed in a body of in vitro studies to be active in limiting infection by a range of different RNA viruses [10,14], including HIV-1 [7], DENV (all four serotypes) and related flaviviruses [8,11,12], influenza, and alphaviruses such as Venezuelan equine encephalitis virus (VEEV) and chikungunya [9,15,16] (see Table 1); it is also active against DNA viruses [18,19,20]. Recent studies indicate it is a potent inhibitor of SARS-CoV-2 [17].

A striking aspect of this antiviral activity is that, where determined, the EC_50_ for viral inhibition as assessed by a range of different techniques is in the low μM range (see right column, Table 1), interestingly aligning perfectly with its activity in inhibiting recognition of viral nuclear import cargoes by IMPα (see top of left column, Table 1). The clear implication is that the mechanism of inhibition of infectious virus production in the case of all of the viruses listed in Table 1 is largely through targeting IMPα to prevent its role in nuclear import, and of viral proteins in particular (see Figure 1). Significantly, two other small molecules (GW5074 and gossypol) that appear to target IMPα in a very similar way to prevent its nuclear import function [29] have comparable antiviral properties [13,29,36], consistent with the idea that the host protein IMPα is a key contributor to infection by a number of medically important viruses.

## 5. Ivermectin as an Antiviral in the Clinic

One of the greatest challenges in antiviral research, as in many other disciplines, is to transition from laboratory experiments to preclinical/clinical studies, with the question of dosing challenging [6]. However, it is important to stress the obvious in this context: that the antiviral activities of ivermectin documented in Table 1 have been derived from laboratory experiments that largely involve high, generally non-physiological, multiplicities of infection, and cell monolayer cultures, often of cell lines such as Vero cells (African green monkey kidney, impaired in interferon α/β production) that are not clinically relevant. Clearly, the results in Table 1 for low μM EC_50_ values should not be interpreted beyond the fact that they reveal robust, dose-dependent antiviral activity in the cell model system used, and it would be naïve to strive for μM concentrations of ivermectin in the clinic based on them.

A key consideration in any clinical intervention using ivermectin is its host-directed (IMPα-directed) mechanism of action. Host-directed agents that impact cellular activities that are essential to healthy function must be tested with caution; although ivermectin has an established safety profile in humans [23,25], and is US Food and Drug Administration-approved for a number of parasitic infections [1,3,5], it targets a host function that is unquestionably important in the antiviral response, and titration of a large proportion of the IMPα repertoire of a cell/tissue/organ likely to lead to toxicity. With this in mind, where a host-directed agent can be a “game-changer” in treating viral infection may well be in the initial stages of infection or even prophylactically (see Section 6) to keep the viral load low so that the body’s immune system has an opportunity to mount a full antiviral response [11,17].

Ivermectin’s real potential as an antiviral to treat infection can, of course, only be demonstrated in preclinical/clinical studies. Preclinical studies include a lethal Pseudorabies (PRV) mouse challenge model which showed that dosing (0.2 mg/kg) 12 h post-infection protected 50% of mice, which could be increased to 60% by administering ivermectin at the time of infection [18]. Apart from the many clinical trials currently running for SARS-CoV-2 (see below), the only other study thus far reported relates to a phase III trial for DENV infection [40]. Almost 70% of the world’s population in over 120 countries is currently threatened by mosquito-borne flaviviral infections, with an estimated 100 million symptomatic DENV infections and up to 25,000 deaths each year from dengue haemorrhagic fever [41,42], despite sophisticated large-scale vector control programs. As for the closely related ZIKV (cause of large outbreaks in the Americas in 2015/2016), the dearth of antiviral treatments and challenges in developing efficacious vaccines hamper disease control. Clinical data published in preliminary form for the phase III trial in Thailand [40] indicate antiviral activity; daily dosing (0.4 mg/kg) was concluded to be safe, and have virological efficacy, but clear clinical benefit was not reported, potentially due to the timing of the intervention. The authors concluded that dosing regimen modification was required to ensure clinical benefit [40]. This study both underlines ivermectin’s potential to reduce viral load in a clinical context, and highlights the complexities of timely intervention and effective dosing regimens to achieve real clinical benefit in the field.

## 6. A Viable Treatment for SARS-CoV-2?

Despite efforts in multiple domains, the current SARS-CoV-2 pandemic has now eclipsed the porcine flu epidemic in terms of numbers of infections (rapidly nearing 30 million) and deaths (>930,000) worldwide. The search for antivirals for SARS-CoV-2 through repurposing existing drugs has proved challenging (e.g., see [43,44,45,46,47]), one important aspect of repurposing being the perceived need to achieve therapeutic levels in the lung. Published pharmacokinetic modelling based on both the levels of ivermectin achievable in human serum from standard parasitic treatment dosing and robust large animal experiments where lung levels of ivermectin can be measured, indicates that concentrations of ivermectin 10 times higher than the c. 2.5 μM EC_50_ indicated by in vitro experiments (Table 1) are likely achievable in the lung in the case of SARS-CoV-2 [48]; modelling based on different assumptions predicts lower values, but stresses the long-term stability of ivermectin in the lung (for over 30 days) based on data from animals [49]. It should also be noted that liquid formulations for intravenous administration of long-acting ivermectin have been described, with aerosol administration also in development, to enable ivermectin administration to achieve even higher concentrations to tackle SARS-CoV-2, whilst the use of ivermectin in combination with other agents may enhance efficacy at lower doses.

There are currently more than 50 trials worldwide testing the clinical benefit of ivermectin to treat or prevent SARS-CoV-2 (see Table 2). 

These include variations on combination therapies (see [51,52]), dosing regimens, and prophylactic protocols. With respect to the latter, preliminary results from recently completed study NCT04422561 (Table 2, no. 22), that examines asymptomatic family close contacts of confirmed COVID patients, show that two doses of ivermectin 72 h apart result in only 7.4% of 203 subjects reporting symptoms of SARS-CoV-2 infection, in stark contrast to control untreated subjects, of whom 58.4% reported symptoms, underlining ivermectin’s potential as a prophylactic. It is to be hoped that the results from the rigorous randomised clinical trials listed in Table 2 will emerge in the next few months to document ivermectin’s credentials as “the real deal” for COVID-19 infection or otherwise. In this context, it is noteworthy that ivermectin has already been approved for the treatment of SARS-CoV-2 in humans by the Republic of Peru [53] and in the Northeastern Beni region of Bolivia [54].

## 7. Conclusions

An instinctive response in developing antiviral agents is to strive for high specificity, making the idea of virus-targeted agents specific to a particular viral component or function attractive, since, ideally, they circumvent the possibility of impacting host function. However, the high propensity of viral genomes, and particularly those of RNA viruses, to mutate and evolve means that selection for resistance can be all too prevalent (e.g., for HIV). Importantly, the high specificity of an agent to a particular virus also inevitably means that its utility against a distinct virus may be limited or non-existent. Thus, it is not surprising that repurposed antivirals active against influenza or HIV, for example, may prove in efficacious against distantly related flaviviruses or coronaviruses.

In contrast, antivirals that are host-directed can be repurposed more easily, as long as the viruses in question rely on the same host pathway/functions for robust infection, simply because the host pathway/function targeted is the same [14,28]. Although potential complication here is the viral tissue tropism (e.g., blood or lung in the case of systemic or respiratory viral infections) and accompanying pharmacokinetic considerations, selection for viral resistance is largely circumvented in this scenario. As long as toxicity is not an issue, host-directed agents thus have the potential to be genuinely broad-spectrum agents against various different viruses that rely on a common host pathway. The fact that so many viruses rely on IMPα/β1-dependent nuclear import for robust infection ([14,27,28] and see above) means that agents targeting this pathway have true potential to be broad-spectrum antivirals. After decades of use in the field, ivermectin clearly “fits the bill” here in terms of human safety, but whether it turns out to be the molecule that proves this principle will only begin to be established unequivocally, one way or another, in the ensuing months with respect to SARS-CoV-2.

## Figures and Tables

**Figure 1 cells-09-02100-f001:**
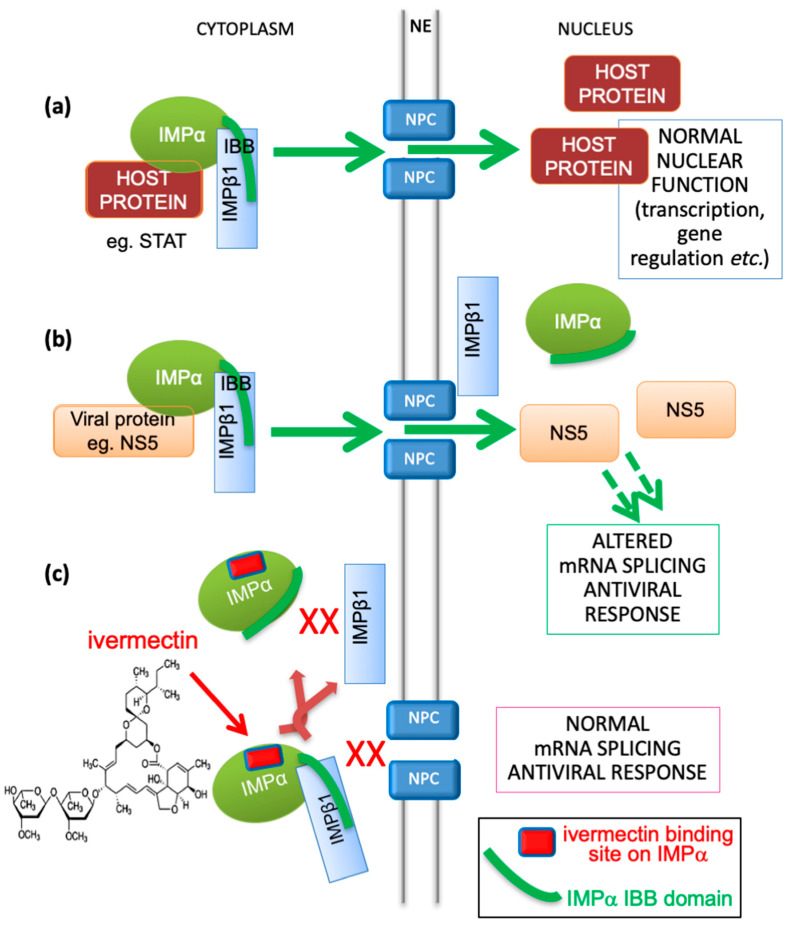
Schematic showing IMPα’s role in nuclear transport of host and viral proteins, and mechanism of inhibition by ivermectin. (**a**) Host proteins, such as members of the STAT or NF-κB transcription factor families, localize in the nucleus through the action of the IMPα/β1 heterodimer, where the “IBB” (IMPβ-binding) region of IMPα (green curved line) is bound by IMPβ1 to enable cargo recognition by IMPα within the heterodimer; IMPβ1 subsequently mediates transport of the trimeric complex through the nuclear pore (NPC, nuclear pore complex) embedded within the nuclear envelope (NE) into the nucleus. This is followed by release within the nucleus to enable the transcription factors to carry out normal function in transcriptional regulation, including in the antiviral response. IMPα can only mediate nuclear import within the heterodimer with IMPβ1. (**b**) In viral infection, specific viral proteins (e.g., NS5 in the case of DENV, ZIKV, WNV) able to interact with IMPα utilize the IMPα/β1 heterodimer to access the nucleus and antagonize the antiviral response [14,27,28]. This is critical to enable optimal virus production as shown by mutagenic and inhibitor studies. Which SARS-CoV-2 proteins may access the nucleus in infected cells has not been examined (see Section 3). (**c**) The IMPα targeting compound ivermectin binds to IMPα (binding site shown as red lozenge) both within the IMPα/β heterodimer to dissociate it, and to free IMPα to prevent it binding to IMPβ1, thereby blocking NS5 nuclear import [11]. GW5074 (see Table 1) has been shown to exhibit a similar mechanism [29].

**Table 1 cells-09-02100-t001:** In vitro properties of IMPα inhibitors with antiviral effects.

Compound	Documented Action on IMPα ^3^	Antiviral Against ^4^	Inhibitory Concentration (Assay)/Fold reduction ^5^
Ivermectin ^1^	Inhibits interaction in vitro of IMPα with HIV-IN [34], DENV2 NS5 (1 μM) [7,11], T-ag [31], Hendra V (15 μM) [13], IMPβ1 (7 μM) [11] Inhibits interaction of IMPα with T-ag and NS5 in a cell context as visualised by quantitative BiFc [11] Inhibits CoIP from cell lysates of IMPα with T-ag, Adenovirus EIA [20] Inhibits nuclear accumulation in a cellular context of IMPα/β1- but not β1-recognised viral proteins such as T-ag [7,16], DENV2 NS5 [7], VEEV Capsid [16], adenovirus E1A [20], PSV UL42 [18] as well as host cargoes (see [15,30]) Reduces nuclear localisation in infected cells of VEEV Capsid [9] and adenovirus E1A [20]	Coronavirus SARS-CoV-2 HIV-1 (VSV-G-pseudotyped NL4-3.Luc.R-E-HIV) Influenza VLPs (avian influenza A/MxA escape mutants) Flavivirus: YFV (17D) DENV1 (EDEN-1) DENV2 (NGC) DENV2 (EDEN-2) DENV3 (EDEN-3) DENV4 (EDEN-4) WNV (NY99) WNV (MRM61C) ZIKV (Asian/Cook Islands/2014) Alphavirus: Chikungunya virus (CHIKV-Rluc) Sindbis (HR) Semliki forest virus VEEV (TC83) Hendra (Hendra virus/Australia/Horse/1994) DNA viruses Adenovirus (HAdV-C5) Adenovirus (HAdV-B3) BK polyomavirus (BKPyV) Pseudorabies	EC_50_ = 2.2/2.8 μM (qPCR/released/cell-associated virus) [17] 5000-fold 50 μM > 2-fold (luciferase) [7] 10 μM total inhibition (luciferase) [10] EC_50_ = 5/0.5 nM (CPE/qPCR) [12] 3 μM > 50,000-fold (pfu) [15] EC_50_ = 2.3/3 μM (CFI, 2 hosts) [8] EC_50_ = 0.7 μM (qPCR) [12] EC_50_ = 0.4/0.6 μM (pfu/qPCR) [11] 50 μM total inhibition (pfu) [7] EC_50_ = 2.1/1.7 μM (CFI, 2 hosts) [8] EC_50_ = 1.7 μM (CFI) [8] EC_50_ = 1.9 μM (CFI) [8] EC_50_ = 4 μM (qPCR) [12] EC_50_ = 1/0.5 μM (pfu/qPCR) [11] EC_50_ = 1.3/1.6 μM (pfu/qPCR) [11] EC_50_ = 1.9/0.6 μM (luciferase, 2 hosts) 3 μM > 5000-fold (pfu) [15] 3 μM > 1000-fold (pfu) [15] 3 μM > 200-fold (pfu) [15] 1 μM c. 20-fold (pfu) [9] est. EC_50_ = 2 μM (TCID/luciferase) [13] EC_50_ = c. 2.5 μM; 10 μM 20-fold (qPCR) [20] 10 μM c. 8-fold (qPCR) [20] Est. EC_50_ 1.5 μM (PFU/CPE/qPCR) [19] Est. EC_50_ c. 0.8 μM 1000-fold [18]
Gossypol ^2^	Inhibits interaction in vitro of IMPα with Hendra Virus V (10 μM) [14] Inhibits nuclear accumulation in WNV infected cells of NS5 [36]	WNV (MRM61C) Hendra (Hendra virus/Australia/Horse/1994)	10 μM 100-fold (pfu) [36] 10 μM (TCID/luciferase) [14]
GW5074	Inhibits interaction in vitro of IMPα with DENV2 NS5 (5 μM) [29], Hendra (V 15 uM) [13], IMPβ1 (10 μM) [29] Inhibits nuclear accumulation in DENV2 infected cells of NS5 [29]	DENV2 NGC ZIKV (Asian/Cook Islands/2014) WNV (MRM61C)	EC_50_ = 0.5/1 (pfu/PCR) [29] EC_50_ = 0.3/0.6 (pfu/PCR) [29] EC_50_ = 5/7 (pfu/PCR) [29]

^1^ US Food and Drug Administration (FDA)-approved broad-spectrum antiparasitic agent, including against parasitic worm infestations and ectoparasites causing scabies, pediulosis and rosacea [1,3]. ^2^ Abandoned as human male contraceptive due to side effects [38,39]. ^3^ Inhibits helicase activity (FRET based assay) of DENV2, YFV and WNV NS3 (IC_50_ 0.2–0.5 μM) [12]. ^4^ Entries in brackets indicate virus strains/constructs used. ^5^ Est. estimated.

**Table 2 cells-09-02100-t002:** Summary of current clinical trials using ivermectin for SARS-CoV-2.

	Title, URL	Status ^1^	N ^2^	Interventions ^3^	Start	Locations
1	Ivermectin Effect on SARS-CoV-2 Replication in Patients With COVID-19; https://ClinicalTrials.gov/show/NCT04381884	R	45	Ivermectin 0.6 mg/kg QD plus SC vs. SC	18.5.20	CEMIC, Buenos Aires, Ciudad De Buenos Aires, Argentina
2	Ivermectin and Nitazoxanide Combination Therapy for COVID-19; https://ClinicalTrials.gov/show/NCT04360356	NY	100	Ivermectin 0.2 mg/kg once plus NZX 500 mg BID for 6 days vs. SC	20.5.20	Tanta University, Egypt
3	Ivermectin vs. Placebo for the Treatment of Patients With Mild to Moderate COVID-19; https://ClinicalTrials.gov/show/NCT04429711	R	100	Ivermectin 12–15 mg/day for 3 days vs. Placebo	12.5.20	Sheba Medical Center, Ramat-Gan, Israel
4	Hydroxychloroquine and Ivermectin for the Treatment of COVID-19 Infection; https://ClinicalTrials.gov/show/NCT04391127	R	200	Ivermectin 12 mg (<80 kg) or 18 mg (>80 kg) once vs. HCQ 400 mg BID for 1 day then 200 mg BID for 4 days vs. Placebo	4.5.20	Jose Manuel Arreola Guerra, Aguascalientes, Mexico
5	Efficacy of Ivermectin in Adult Patients With Early Stages of COVID-19. https://ClinicalTrials.gov/show/NCT04405843	NY	400	Ivermectin 0.3 mg/kg daily for 5 days vs. Placebo	20.6.20	Colombia
6	Ivermectin In Treatment of COVID 19 Patients. https://ClinicalTrials.gov/show/NCT04425707	R	100	Ivermectin (dose unlisted) vs. SC vs. Ivermectin (dose unlisted) plus SC	9.6.20	Isolation and referral hospitals for COVID 19 patients, Cairo, Egypt
7	Efficacy and Safety of Ivermectin and Doxycycline in Combination or IVE Alone in Patients With COVID-19 Infection; https://ClinicalTrials.gov/show/NCT04407130	E	72	Ivermectin 0.2 mg/kg once plus 200 mg DOC day 1 followed by 100 mg DOC BID for 4 days vs. Ivermectin 0.2 mg/kg QD for 5 days vs. Placebo	16.6.20	Icddr, B, Dhaka, Bangladesh
8	Efficacy of Ivermectin as Add on Therapy in COVID19 Patients. https://ClinicalTrials.gov/show/NCT04343092	C	100	Ivermectin 0.2 mg/kg once weekly plus HCQ 400 mg QD plus ATM 500 mg QD vs. HCQ 400 mg QD plus ATM 500 mg QD	18.4.20	General Directorate of Medical City, Bagdad, Baghdad, Iraq
9	COVidIVERmectin: Ivermectin for Treatment of Covid-19 (COVER). https://ClinicalTrials.gov/show/NCT04438850	NY	102	Ivermectin 0.6 mg/kg QD for 5 days vs. Ivermectin 1.2 mg/kg QD for 5 days vs. Placebo	20.6.20	Negrar, Verona, Italy; Bologna, Italy; Milan, Italy; Rovereto, Italy; Turin, Italy; Barcelona, Spain; Madrid, Spain
10	Efficacy, Safety and Tolerability of Ivermectin in Subjects Infected With SARS-CoV-2 With or Without Symptoms (SILVERBULLET). https://ClinicalTrials.gov/show/NCT04407507	NY	66	Ivermectin 12 mg/day for 3 days plus paracetamol 500 mg QID for 14 days vs. Placebo plus paracetamol 500 mg QID for 14 days	20.6.20	Investigacion Biomedica para el Desarrollo de Farmacos S.A. de C.V., Mexico
11	Sars-CoV-2/COVID-19 Ivermectin Navarra-ISGlobal Trial (SAINT). https://ClinicalTrials.gov/show/NCT04390022	R	24	Ivermectin 0.4 mg/kg once vs. Placebo	14.5.20	Clinica Universidad de Navarra, Pamplona, Navarra, Spain
12	A Comparative Study on Ivermectin and Hydroxychloroquine on the COVID19 Patients in Bangladesh. https://ClinicalTrials.gov/show/NCT04434144	C	116	Ivermectin 0.2 mg/kg once plus DOC 100 mg BID for 10 days vs. HCQ 400 mg day 1 then 200 mg BID for 9 days plus ATM 500 mg/day for 5 days	2.5.20	Chakoria Upazilla Health Complex, Cox’s Bazar, Bangladesh
13	Ivermectin vs Combined Hydroxychloroquine and Antiretroviral Drugs (ART) Among Asymptomatic COVID-19 Infection (IDRA-COVID19). https://ClinicalTrials.gov/show/NCT04435587	NY	80	Ivermectin 0,6 mg/kg daily for 3 days vs. HCQ 400 mg BID Day 1 then 200 mg BID for 4 days plus Darunavir/ritonavir (400 mg/100 mg) BID for 5 days	20.7.20	Siriraj Hospital, Bangkok Noi, Bangkok, Thailand
14	IVERMECTIN Aspirin Dexametasone and Enoxaparin as Treatment of Covid 19. https://ClinicalTrials.gov/show/NCT04425863	A	100	Ivermectin 5 mg/mL oral to be repeated 1 week later (dose unlisted)	1.5.20	Hospital Eurnekian, Buenos Aires, Argentina
15	A Preventive Treatment for Migrant Workers at High-risk of Covid-19. https://ClinicalTrials.gov/show/NCT04446104	R	5000	Ivermectin 12 mg once vs. HCQ 400 mg day 1 then 200 mg/day for 42 days vs. Zinc 80 mg/day plus vitamin C 500 mg/day for 42 days vs. Povidone-iodine throat spray TID for 42 days vs. Vitamin C 500 mg/day for 42 days	13.5.20	Tuas South Dormitory, Singapore, Singapore
16	New Antiviral Drugs for Treatment of COVID-19. https://ClinicalTrials.gov/show/NCT04392427	NY	100	Ivermectin (dose unlisted) plus NZX (dose unlisted) plus ribavirin 200 mg or 400 mg vs. Control (untreated)	20.5.20	Mansoura University, Mansoura, Select A State Or Province, Egypt
17	Early Treatment With Ivermectin and LosarTAN for Cancer Patients With COVID-19 Infection (TITAN). https://ClinicalTrials.gov/show/NCT04447235	NY	176	Ivermectin 12 mg once plus losartan 50 mg/day for 15 days vs. Placebo	20.7.20	Instituto do Cancer do Estado de SÃ£o Paulo, Brazil
18	Ivermectin in Treatment of COVID-19. https://ClinicalTrials.gov/show/NCT04445311	R	100	Ivermectin daily (dose unlisted) for 3 days plus SC vs. SC	31.5.20	Waheed Shouman, Zagazig, Sharkia, Egypt
19	Efficacy of Ivermectin in COVID-19. https://ClinicalTrials.gov/show/NCT04392713	R	100	Ivermectin 12 mg once plus SC vs. SC	15.4.20	Combined Military Hospital Lahore, Lahore, Punjab, Pakistan
20	Ivermectin and Doxycycine in COVID-19 Treatment. https://ClinicalTrials.gov/show/NCT04403555	NY	40	Ivermectin (dose unlisted) plus DOC (dose unlisted) vs. CQ (dose unlisted)	1.6.20	Sherief Abd-Elsalam, Tanta, Egypt
21	The Efficacy of Ivermectin and Nitazoxanide in COVID-19 Treatment. https://ClinicalTrials.gov/show/NCT04351347	R	300	Ivermectin (dose unlisted) vs. Ivermectin (dose unlisted) plus NZX (dose unlisted) vs. Ivermectin (dose unlisted) plus CQ (dose unlisted)	16.6.20	Tanta University, Tanta, Egypt
22	Prophylactic Ivermectin in COVID-19 Contacts. https://ClinicalTrials.gov/show/NCT04422561	C ^4^	304	Ivermectin 15 mg (40–60 kg), 18 mg (60–80 kg) or 24 mg (> 80 kg) per day, 2 doses 72 h apart vs. Control (untreated)	31.5.20	Zagazig University, Zagazig, Sharkia, Egypt
23	Max Ivermectin- COVID 19 Study Versus Standard of Care Treatment for COVID 19 Cases. A Pilot Study. https://ClinicalTrials.gov/show/NCT04373824	R	50	Ivermectin 0.2 mg/kg daily for 2 days plus SC vs. SC	25.4.20	Max Super Speciality hospital, Saket (A unit of Devki Devi Foundation), New Delhi, Delhi, India
24	A Study to Compare the Efficacy and Safety of Different Doses of Ivermectin for COVID-19 (IFORS) https://ClinicalTrials.gov/show/NCT04431466	NY	64	Ivermectin 0.1 mg/kg once vs. Ivermectin 0.1 mg/kg day 1 and repeated after 72 h vs. Ivermectin 0.2 m/kg once vs. Ivermectin 0.2 mg/kg day 1 and repeated after 72 h vs. SC	1.7.20	Hospital Univeristário da Universidade Federal de São Carlos (HU-UFSCar), São Carlos, São Paulo, Brazil
25	Novel Agents for Treatment of High-risk COVID-19 Positive Patients https://ClinicalTrials.gov/show/NCT04374019	R	240	Ivermectin 12 mg (<75 kg) or 15 mg (>75 kg) daily for 2 days vs. HCQ 600 mg/day for 14 days plus ATM 500 mg day 1 then 250 mg/day for 4 days vs. Camostat Mesilate 200 mg TID for 14 days vs. Artemesia annua 50 mg TID for 14 days	1.5.20	University of Kentucky Markey Cancer Center, Lexington, Kentucky, United States
26	Ivermectin-Azithromycin-Cholecalciferol (IvAzCol) Combination Therapy for COVID-19 (IvAzCol) https://ClinicalTrials.gov/show/NCT04399746	R	30	Ivermectin 6 mg/day on days 0, 1, 7 and 8 plus ATM 500 mg/day 4 days plus Cholecalciferol 400 IU BID for 30 days vs. Control (untreated)	15.3.20	Outpatient treatment, Mexico City, Mexico
27	USEFULNESS of Topic Ivermectin and Carrageenan to Prevent Contagion of Covid 19 (IVERCAR) https://ClinicalTrials.gov/show/NCT04425850	A ^5^	1195	Ivermectin (topical for oral mucosae) plus iota carrageenan (topical for oral mucosae) 5 times per day plus PPE vs. PPE only	1.6.20	Hospital Eurnekian, Buenos Aires, Argentina
28	Novel Regimens in COVID-19 Treatment https://ClinicalTrials.gov/show/NCT04382846	NY	80	Ivermectin plus CQ (dose unlisted) vs. Ivermectin plus NZX (dose unlisted) vs. Ivermectin plus NZX plus ATM (dose unlisted) vs. NZX and ATM (dose unlisted)	8.5.20	Tanta University, Egypt
29	Anti-Androgen Treatment for COVID-19 https://ClinicalTrials.gov/show/NCT04446429	NY	254	Ivermectin 0.2 mg/kg QD plus ATM 500 mg QD vs. Ivermectin 0.2 mg/kg QD plus ATM 500 mg QD plus Dutasteride 0.5 mg QD	26.6.20	Corpometria Institute, Brasilia, Brazil
30	A Real-life Experience on Treatment of Patients With COVID 19 https://ClinicalTrials.gov/show/NCT04345419	R	120	Ivermectin (dose unlisted) vs. CQ (dose unlisted) vs. Favipiravir (dose unlisted) vs. NZX (dose unlisted) vs. Niclosamide (dose unlisted) vs. other drugs (oseltamivir or combination of above, dose unlisted)	16.6.20	Tanta university hospital, Tanta, Egypt
31	Worldwide Trends on COVID-19 Research After the Declaration of COVID-19 Pandemic (observational) https://ClinicalTrials.gov/show/NCT04460547	NY	200	Completed interventional vs. completed observational studies on Ivermectin, Convalescent Plasma, HCQ, DAS181, or Interferon 1A	25.7.20	Qassim University, Saudi Arabia
32	Trial of Combination Therapy to Treat COVID-19 Infection https://ClinicalTrials.gov/show/NCT04482686	NY	300	Ivermectin (dose unlisted) day 1 and 4 plus DOC (dose unlisted) for 10 days plus Zinc for 10 days plus Vitamin D3 for 10 days plus Vitamin C for 10 days vs. Placebo	22.7.20	ProgenaBiome, California, USA
33	Randomised clinical trial of ivermectin for treatment and prophylaxis of COVID-19 https://www.clinicaltrialsregister.eu/ctr-search/trial/2020-001994-66/ES	O	266	Ivermectin (dose unlisted) vs. Placebo	8.5.20	Fundació Assistencial Mútua Terrassa, Spain
34	Multicenter, randomized, double-blind, placebo-controlled study investigating efficacy, safety and tolerability of ivermectin HUVE-19 in patients with proven SARS-CoV-2 infection (COVID-19) and manifested clinical symptoms. https://www.clinicaltrialsregister.eu/ctr-search/trial/2020-002091-12/BG	O	120	Ivermectin 0.4 mg/kg plus SC vs. Placebo plus SC	5.5.20	Bulgaria (9 sites)
35	Efficacy of hydroxychloroquine, ciclesonide and ivermectin in treatment of moderate covid-19 illness: an open-label randomised controlled study (EHYCIVER-COVID) http://ctri.nic.in/ClinicaltrialsCTRI/2020/04/024948	NY	120	Ivermectin 12 mg/day for 7 days vs. Ciclesonide 0.2 mg/kg BID for 7 days vs. HCQ 400 mg BID Day 1 then 200 mg BID for 6 days vs. SC	15.5.20	New Delhi, India
36	A Phase IIB open label randomized controlled trial to evaluate the efficacy and safety of Ivermectin in reducing viral loads in patients with hematological disorders who are admitted with COVID 19 infection http://ctri.nic.in/ClinicaltrialsCTRI/2020/04/025068	NY	50	Ivermectin 3 mg (15–24 kg) or 6 mg (25–35 kg) or 9 mg (36–50 kg) or 12 mg (51–65 kg) or 15 mg (66–79 kg) or 0.2 mg/kg (80 kg) once vs. SC	27.5.20	Christian Medical College Vellore, TAMIL NADU, India
37	Interventional study to assess the efficacy of Ivermectin with standard of care treatment versus standard of care in patients of COVID-19 at R D Gardi Medical College, Ujjain, India http://ctri.nic.in/ClinicaltrialsCTRI/2020/04/025224	NY	50	Ivermectin 12 mg/day for 2 days plus SC vs. SC	24.5.20	R D Gardi Medical College, Ujjain, Madhya Pradesh, India
38	Study to assess the efficacy of Ivermectin as prophylaxis of COVID 19 among health care workers and COVID 19 contacts in Ujjain, India; http://ctri.nic.in/ClinicaltrialsCTRI/2020/04/025333	NY	2000	Ivermectin 12 mg/day (adult) or 6 mg/day (children) for 2 days vs. Control	27.5.20	R D Gardi Medical College, Ujjain, Madhya Pradesh, India
39	Randomised Controlled Trial of Ivermectin in hospitalised patients with COVID19 (RIVET-COV). http://ctri.nic.in/ClinicaltrialsCTRI/2020/04/026001	NY	60	Ivermectin single dosing of 0.2 mg/kg vs. Ivermectin 0.4 mg/kg vs. Ivermectin 0.8 mg/kg vs. Ivermectin 1.6 mg/kg vs. Ivermectin 2 mg/kg vs. SC	25.6.20	New Delhi, India
40	A Prospective, randomized, single centred, open labelled, two arm, placebo-controlled trial to evaluate efficacy and safety of Ivermectin drug in patients infected with SARS-CoV-2 virus; http://ctri.nic.in/ClinicaltrialsCTRI/2020/04/025960	NY	100	Ivermectin 12 mg/day for 3 days vs. SC	18.6.20	Symbiosis University Hospital and Research Centre, Maharashtra, India
41	A Clinical Trial to Study the Efficacy of “Ivermectin” in the prevention of Covid-19. A Single Arm Study. http://ctri.nic.in/ClinicaltrialsCTRI/2020/04/026232	NY	50	Ivermectin 0.2 mg/kg once	10.7.20	DVFM, Andhra Pradesh, India
42	Ivermectin Nasal Spray for COVID19 Patients. https://ClinicalTrials.gov/show/NCT04510233	NY	60	Ivermectin nasal spray (1 mL) in each nostril BID vs. Ivermectin oral (6 mg) TID vs. SC	10.8.20	Tanta University, Tanta, Egypt
43	Outpatient use of ivermectin in COVID-19. https://ClinicalTrials.gov/show/NCT04530474	NY	200	Ivermectin 0.15–0.2 mg/kg (max 12 mg) once vs. Placebo	26.8.20	Temple University Hospital, Philadelphia, USA
44	Ivermectin to prevent hospitilizations in COIVD-19. https://ClinicalTrials.gov/show/NCT04529525	R	500	Ivermectin 12 mg (48–80 kg) or 18 mg (80–110 kg) or 24 mg (>100 kg) at inclusion and again at 24h vs. Placebo	21.8.20	Ministry of Public Health, Province of Corrientes, Argentina
45	Clinical trial of ivermectin plus doxycycline for the treatment of confirmed Covid-19 infection. https://ClinicalTrials.gov/show/NCT04523831	R	400	Ivermectin 6 mg and doxycycline 100 mg BID for 5 days vs. Placebo	19.8.20	Dhaka Medical College, Dhaka Bangladesh
46	Pilot study to evaluate the potential of ivermectin to reduce COVID-19 transmission. https://www.clinicaltrialsregister.eu/ctr-search/trial/2020-001474-29/ES	O	24	Ivermectin (dose unlisted) vs. Placebo	8.5.20	Clinica Universidad de Navarra, Pamplona, Spain
47	Dose-Finding study of Ivermectin treatment on patients infected with Covid-19: A clinical trial. https://en.irct.ir/trial/47012	A	125	Ivermectin 0.2 mg/kg single dose plus SC vs. Ivermectin 0.2 mg/kg day 1, 2, 5 plus SC vs. Placebo plus SC vs. Ivermectin 0.4 mg/kg day 1 and 0.2 mg/kg day 2, 5 vs. SC	4.5.20	Qazvin University of Medical Sciences, Qazvin, Iran
48	In vivo use of ivermectin (IVR) for treatment for corona virus infected patients: a randomized controlled trial. http://www.chictr.org.cn/showprojen.aspx?proj=54707	NY	60	Ivermectin single dose 0.2 mg/kg vs. Placebo	10.6.20	Rayak Hospital, Riyaq, Lebanon
49	A randomized clinical trial study, comparison of the therapeutic effects of Ivermectin, Kaletra and Chloroquine with Kaletra and Chloroquine in the treatment of patients with coronavirus 2019 (COVID-19). http://en.irct.ir/trial/48444	A	60	Ivermectin 0.15–0.2 mg/kg single dose day 1 plus HCQ 200 mg day 1 plus Lopinavir/Ritonavir 400/100 mg days 2–6 vs. HCQ 200 mg day 1 plus Lopinavir/Ritonavir 400/100 mg days 2-6	30.5.20	Ahvaz Razi Hospital, Ahvaz, Iran
50	A double-blind clinical trial to repurpose and assess the efficacy and safety of ivermectin in COVID-19. http://isrctn.com/ISRCTN40302986	R	45	Ivermectin 6 mg every 3.5 days for 2 weeks vs. Ivermectin 12 mg every 3.5 days for 2 weeks vs. Placebo	23.4.20	Lagos University Teaching Hospital, Lagos, Nigeria
51	Effectiveness of Ivermectin in the Treatment of Coronavirus Infection in Patients admitted to Educational Hospitals of Mazandaran in 2020. https://en.irct.ir/trial/49174	R	60	Ivermectin 0.2 mg/kg once plus SC vs. SC	21.5.20	Bouali Hospital, Sari, Iran
52	Subcutaneous Ivermectin in Combination With and Without Oral Zinc and Nigella Sativa: a Placebo Randomized Control Trial on Mild to Moderate COVID-19 Patients. https://clinicaltrials.gov/ct2/show/study/NCT04472585	R	40	Ivermectin 0.2 mg/kg subcutaneous injection every 2 days plus SC vs. Ivermectin 0.2 mg/kg subcutaneous injection every 2 days plus 80 mg/kg Nigella Sativa oral QD plus SC vs. Ivermectin 0.2 mg/kg subcutaneous injection every 2 days plus 20 mg Zinc Sulfate oral TID plus SC vs. Placebo plus SC	14.7.20	Shaikh Zayed Hospital, Lahore, Pakistan
53	Pragmatic study “CORIVER”: Ivermectin as antiviral treatment for patients infected by SARS-COV2 (COVID-19). https://www.clinicaltrialsregister.eu/ctr-search/trial/2020-001971-33/ES	O	45	Ivermectin 0.2–0.4 mg/kg (regime unlisted) vs. HCQ 400 mg vs. ATM 500 mg vs. Placebo	22.7.20	Hospital Universitario Virgen de las Nieves, Granada, Spain
54	Effectiveness and Safety of Ivermectin for the Prevention of Covid-19 Infection in Colombian Health Personnel at All Levels of Care, During the 2020 Pandemic: A Randomized Clinical Controled Trial. https://clinicaltrials.gov/ct2/show/record/NCT04527211	NY	550	Ivermectin 0.2 mg/kg weekly for 7 weeks vs. Placebo	7.9.20	Pontificia Universidad Javeriana, Valle Del Cauca, Colombia

^1^ R, Recruiting, NY, Not yet recruiting, A, Active not recruiting, C, Completed, E, Enrolling by invitation, O, ongoing ^2^ Number of patients ^3^ SC, standard care, QD, once per day, BID, twice daily, QID, 4 times daily, TID, 3 times daily, PPE, personal protective equipment, vs. versus, HCQ, hydroxychloroquine (US Food and Drug Administration approval was rescinded for COVID-19) [44,45]; DOC, doxycycline; CQ, chloroquine, ATM, Azithromycin, NZX, Nanozoxide ^4^ Raw data for asymptomatic family close contacts of confirmed COVID patients show that 2 doses of ivermectin 72 h apart resulted in only 7.4% of 203 subjects reporting symptoms of SARS-CoV-2 infection, in contrast to control untreated subjects, of whom 58.4% reported symptoms; evidence of prophylaxis. ^5^ Preliminary results for 1195 subjects are consistent with prophylaxis [50].

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
