# Peer review of "Ivermectin as a Broad-Spectrum Host-Directed Antiviral: The Real Deal?"

_cells, 2020, doi:10.3390/cells9092100_

Round 1
Reviewer 1 Report
The review article 'Ivermectin as a broad spectrum host-directed antival: the real deal?’ by Jans and Wagstaff describes the action of the anti-parasitic drug, ivermectin, against a number of viral infections, particularly SARS-CoV-2. The review discusses the action of invermectin which has pharmacological activity though binding to IMPa and preventing nuclear trafficking of key proteins. The manuscript highlights that several viruses are also affected through inhibition of nuclear trafficking via IMPa and that ivermectin acts as an antiviral compound against viruses including SARS-CoV-2, DENV and ZIKV.
Given the importance of this topic in the context of SARS-CoV-2, several recent publications have covered the topic of ivermectin activity (particularly the nature review by Heidary and Gharebahi, 2020; Mudatsir et al., 2020; Sharun et al., 2020); however, this study also looks at other important viruses and therefore adds some knowledge as to the activity of ivermectin as an antiviral compound.
There are a few points that the authors should address in order to improve their manuscript.
General comments:
General proof reading is required to improve the English grammar and particularly the punctuation used.
Specific comments:
The authors should be aware of their use of abbreviations for instance
Line 11- FDA
Line 14- HIV is abbreviated but not dengue or zika etc
Dengue/ zika are abbreviated in the table an thereafter in the main text but not before. This should be corrected.
line 15- add 'has' ('and has since been shown...')
Line 18- add 'virus' after dengue and zika, virus should have a small v after WNV, chikungunya should not be capitalised
Remove additional words from in text citations e.g. Line 45/46 (eg 12)
Line 80- why is Ran not shown in figure 1?
Some information included in the figure legend would be better suited to the main body. e.g. Line 99 IMPa...IMPb1; Line 102-106 This is critical.... [30]. Line 109-111 GW5074....(unpublished).
Line 118: add 'to' ('it was initially shown to inhibit ...')
Table 1 - Semliki forest virus is spelt wrong and v of virus should be lower case. Should make it clear that the brackets after the virus are the strain used. Erroneous symbols in column 1. Some viruses are abbreviated and others aren't. Some abbreviations not explained e.g. pfu, TCID, CFI, PCR etc.
Line 163- It is no longer good practice to use 'unpublished' in a manuscript
Line 166 and 203- Remove space between end of sentence and punctuation
The authors should include a concluding remarks section to finish the review.
The authors should include more information about the other viruses that are shown to be affected by ivermectin. The key words include flaviviruses such as denv and zikv but very little is actually mentioned of them in the main text.
Author Response
Reviewer 1 (responses in italics)
General proof reading is required to improve the English grammar and particularly the punctuation used.
We have proofed the manuscript and thank the reviewer for the alert.
Specific comments:
The authors should be aware of their use of abbreviations for instance
Line 11- FDA
We now use US. Food and Drug Administration throughout.
Line 14- HIV is abbreviated but not dengue or zika etc. Dengue/ zika are abbreviated in the table an thereafter in the main text but not before. This should be corrected.
All abbreviations have been corrected as requested.
line 15- add 'has' ('and has since been shown...')
Corrected.
Line 18- add 'virus' after dengue and zika, virus should have a small v after WNV, chikungunya should not be capitalised
All corrections have been made.
Remove additional words from in text citations e.g. Line 45/46 (eg 12)
All additional words have been removed throughout the manuscript as requested.
Line 80- why is Ran not shown in figure 1?
Ran was not shown for reasons of simplicity – we have deleted the text mentioning it from the body of the text and thank the reviewer for pointing out this potential point of confusion, and making the text much clearer.
Some information included in the figure legend would be better suited to the main body. e.g. Line 99 IMPa...IMPb1;
Line 99 is now encapsulated in main text (lines 149-150).
Line 102-106 This is critical.... [30].
Lines 102-106 now encapsulated in the main text (line 90-95) as requested.
Line 109-111 GW5074....(unpublished).
Line 109-111 are now encapsulated in the main text (lines 152-156) as requested.
Line 118: add 'to' ('it was initially shown to inhibit ...')
Added
Table 1 - Semliki forest virus is spelt wrong and v of virus should be lower case. Should make it clear that the brackets after the virus are the strain used. Erroneous symbols in column 1. Some viruses are abbreviated and others aren't. Some abbreviations not explained e.g. pfu, TCID, CFI, PCR etc.
Typographical errors have been amended. We have added a new footnote to explain the brackets. Column one contains two footnotes which are appropriate and no “erroneous symbols”. As indicated above, we have attempted to minimize the use of abbreviations for the sake of clarity, and only used abbreviations (eg. for viruses) where it was necessary to optimize the space in the table format. All required abbreviations for the table are now defined.
Line 163- It is no longer good practice to use 'unpublished' in a manuscript
Deleted
Line 166 and 203- Remove space between end of sentence and punctuation
Spaces removed to close up text as requested.
The authors should include a concluding remarks section to finish the review.
We have added Section 7 as requested – this section returns to the broad spectrum antiviral theme in the title; we thank the Reviewer for this suggestion.
The authors should include more information about the other viruses that are shown to be affected by ivermectin. The key words include flaviviruses such as denv and zikv but very little is actually mentioned of them in the main text.
We have included new text expounding on the world health problems of dengue and zika infection where appropriate, to satisfy the Reviewer. This includes new text (lines 202-207) and accompanying references.
We thank the Reviewer for his/her perspicacious contribution to improving the review and making it more complete.
Reviewer 2 Report
This is an interesting review of published papers on antiviral activity of ivermectin. I found this paper relevant and certainly worth reading. Having principally a clinical and epidemiologic background, I will just discuss those aspects that bear a potential clinical interest. As they are what really matters, I would prompt the authors to discuss them in depth, to help the reader understand if ivermectin really bears a potential as an antiviral, and for COVID-19 in particular.
Major remarks
- Ivermectin has so far shown antiviral activity in vitro against a number of (both RNA and DNA) viruses, and this for several years already. Yet, to my knowledge no convincing proof of clinical activity has been attained, on any of those viruses. Why is this so? The authors should comment on this apparently major contradiction.
- On the same line: the authors correctly cite the only (placebo-controlled) trial that showed antiviral activity in vivo for dengue. The authors, while recognizing that a clinical efficacy could not be demonstrated (possibily due to underdosing), report all the same a clear activity in reducing the viral load. This would be per se a major outcome. Why, I wonder, this finding has just remained confined to a congress abstract, and has never been formally published? This gives rise to some concern.
- It is not clear, to a poor clinician, if and how effective concentrations may be reached in the lung tissue at the currently used dose schedules. What is a “low” uM range for EC50? Low in comparison to what? There is just one paper, among those cited by the authors, that (according to what is stated at lines 172-174) would show that lung tissue concentration 10 times higher than the in vitro EC50 are achievable with the currently approved dose regimens. However, the same paper cited (contradicted by other papers as correctly reported, by the way) shows (Figure 2) that the ratio Cmax/EC90 ratio is particularly low for ivermectin, compared with a lot of other candidates that, however, if tested in clinical trials have failed to meet expectations. Why should then ivermectin work, when other compounds with a seemingly much better profile have failed?
- As is well known, one of the compounds that have (at least so far) failed to show a clinical efficacy is chloroquine (or hydroxichloroquine), despite reaching comparatively much higher plasma concentration at the usual doses, and having a volume of diffusion exceedingly high if compared with ivermectin. Again, why should ivermetin succeed where chloroquine has failed?
- Most importantly. The authors expect a final word (for COVID-19) to come from the planned or ongoing RCTs. I argue that most of these trials are using dose regimens that, to my view, are implausible, and unlikely to have a minimal chance of showing efficacy. Therefore, the statement in line 210 appears rather wishful thinking than reasonable hope.
Minor remarks
- Please check the refs, some of them do not seem to correspond to the subject indicated in the text.
I would really wish the authors to discuss, based on their undisputable knowledge and experience of this drug, the points summarized above, and allow the readers to understand if this (really) wonderful drug has the potential of becoming an antiviral, and a promising drug for COVID-19, first of all.
Thank you for giving me the opportunity to read this really interesting paper, that I hope to see further improved in the final version.
Author Response
Reviewer 2
Major remarks
- Ivermectin has so far shown antiviral activity in vitro against a number of (both RNA and DNA) viruses, and this for several years already. Yet, to my knowledge no convincing proof of clinical activity has been attained, on any of those viruses. Why is this so? The authors should comment on this apparently major contradiction.
The Reviewer is undoubtedly aware that human clinical trials are a very significant financial investment, and this is always by pharmaceutical companies etc. against the “market value” of the health problem itself. Since dengue/zika are seen to be largely third world problems, it has proved difficult to make the business case for trials for these viruses. Rather than point this out, however, we have added additional text, as requested by Reviewer 1, highlighting the health problem and alluding to the other control strategies that have been favoured by most countries (lines 202-207). We thank the Reviewer for pointing this out.
- On the same line: the authors correctly cite the only (placebo-controlled) trial that showed antiviral activity in vivo for dengue. The authors, while recognizing that a clinical efficacy could not be demonstrated (possibily due to underdosing), report all the same a clear activity in reducing the viral load. This would be per se a major outcome. Why, I wonder, this finding has just remained confined to a congress abstract, and has never been formally published? This gives rise to some concern.
Our understanding is that the professor leading this study has retired, and that the remainder of the team is now running clinical trials for ivermectin and SARS-CoV2. We take the Reviewer’s point and now stress more clearly that the trial data is preliminary (line 207). We thank the Reviewer for pointing this out.
- It is not clear, to a poor clinician, if and how effective concentrations may be reached in the lung tissue at the currently used dose schedules. What is a “low” uM range for EC50? Low in comparison to what? There is just one paper, among those cited by the authors, that (according to what is stated at lines 172-174) would show that lung tissue concentration 10 times higher than the in vitro EC50 are achievable with the currently approved dose regimens. However, the same paper cited (contradicted by other papers as correctly reported, by the way) shows (Figure 2) that the ratio Cmax/EC90 ratio is particularly low for ivermectin, compared with a lot of other candidates that, however, if tested in clinical trials have failed to meet expectations. Why should then ivermectin work, when other compounds with a seemingly much better profile have failed?
Because an agent performs in vitro and appears to have appropriate PK properties, does not mean it will work in the human disease – this can only be shown in clinical trials. Ivermectin, in contrast to many other compounds tested that have failed, is a host-targeted treatment that can work to limit the viral load and allow the immune system to function fully to eliminate the viral infection. To satisfy the Reviewer, we now address the question of virus-directed versus host-directed antiviral agents in Section 7 (also requested by Reviewer 1), explaining why it is more challenging to repurpose a virus-directed agent (lines 243-263).
- As is well known, one of the compounds that have (at least so far) failed to show a clinical efficacy is chloroquine (or hydroxichloroquine), despite reaching comparatively much higher plasma concentration at the usual doses, and having a volume of diffusion exceedingly high if compared with ivermectin. Again, why should ivermetin succeed where chloroquine has failed?
See response to 3; in contrast to chloroquine, ivermectin has no toxic side-effects, and is host-directed, rather directed at the virus (in the case of chloroquine, “virus release”). Titrating many, many virus particles to ensure each one is inhibited (chloroquine) is rather different to inhibiting a host agent to a sufficient extent to allow the host to eliminate the virus itself. This was explained in detail in lines 190-195. We do not think it appropriate to criticize work pursuing other agents that are non-specifically virus targeted, but now cite a few clinical trials for these agents (references 43-47).
- Most importantly. The authors expect a final word (for COVID-19) to come from the planned or ongoing RCTs. I argue that most of these trials are using dose regimens that, to my view, are implausible, and unlikely to have a minimal chance of showing efficacy. Therefore, the statement in line 210 appears rather wishful thinking than reasonable hope.
We apologise that the Reviewer has misunderstood our conclusion - RCTs will DEFINITELY give an answer one way or the other – “reasonable hope” is in getting a clear answer, one way or the other. We spell this out clearly, we think in lines 236-239, as well as in the conclusion to section 7 (lines 260-263.
We have expanded Table 2 to include all current running trials, and included mention of the two trials for which there are raw/preliminary results (both quite positive) in addition to citing additional published studies (refs. 51/52). We thank the Reviewer for his/her suggestions.
Minor remarks
- Please check the refs, some of them do not seem to correspond to the subject indicated in the text.
Refs have been checked (yes there were errors) – we thank the Reviewer !
We thank the Reviewer for his/her perspicacious contribution to improving the review and making it more complete. We beg the Reviewer to respect our reluctance to criticize the work of others - we are convinced that in the current situation, definitive (and critically assessed) RCTs are welcome for all po